# Current Management Condition and Waste Composition Characteristics of Construction and Demolition Waste Landfills in Hanoi of Vietnam

Hoang Giang Nguyen [1], Dung Tien Nguyen [2], Ha Tan Nghiem [2], Viet Cuong Tran [1], Akira Kato [3], Akihiro Matsuno [4], Yugo Isobe [5], Mikio Kawasaki [5] and Ken Kawamoto [1,4,*]

1. Faculty of Building and Industrial Construction, Hanoi University of Civil Engineering, No.55 Giai Phong Street, Hai Ba Trung District, Hanoi 11616, Vietnam; giangnh@nuce.edu.vn (H.G.N.); vietcuongx50@gmail.com (V.C.T.)
2. International Cooperation Department, Hanoi University of Civil Engineering, No.55 Giai Phong Street, Hai Ba Trung District, Hanoi 11616, Vietnam; dungnt1@nuce.edu.vn (D.T.N.); tannh@nuce.edu.vn (H.T.N.)
3. Division of Civil Engineering, National Institute of Technology, Tomakomai College, 443 Nishiokka, Tomakomai, Hokkaido 0591275, Japan; a-kato@tomakomai-ct.ac.jp
4. Graduate School of Science and Engineering, Saitama University, 255 Shimo-okubo, Sakura-ku, Saitama 3388570, Japan; matsuno2017@mail.saitama-u.ac.jp
5. Material Cycles and Waste Management Group, Center for Environmental Science in Saitama, 914 Kamitanadare, Kazo City, Saitama 3470015, Japan; isobe.yugo@pref.saitama.lg.jp (Y.I.); kawasaki.mikio@pref.saitama.lg.jp (M.K.)
* Correspondence: kawamoto@mail.saitama-u.ac.jp; Tel.: +81-48-858-3542

**Abstract:** This study collected basic information and conducted waste composition surveys to identify the present management condition of construction and demolition waste (CDW) landfills in Hanoi of Vietnam and to characterize waste composition and grain size distribution of CDW dumped at landfills. Basic information on seven CDW landfills under operation or closed/abandoned was collected, and the waste composition and the grain size distributions of dumped CDW at two landfills were investigated. Results showed that only one landfill site is currently under operation in Hanoi. Sanitary conditions of investigated landfills were relatively good without dumping of hazardous waste. Illegal dumping of domestic waste from residents, however, could be observed more or less at all sites due to an unclear boundary between dumping and surrounding areas. To improve current management of CDW landfills, a suitable recording system of accepted/dumped CDW and technical support for site managers are required as well as the implementation plan for developing and renovating landfills. Based on the results of the waste composition survey, the major components of dumped CDW were "Concrete", "Clay bricks", and "Soil-like", and the sum of these materials reached >80% of the total. Grain size distributions of "Concrete" ranged from 10–600 mm and of "Clay bricks" ranged from 10–300 mm. Technical recommendations to examine a potential availability of dumped "Concrete" and "Clay bricks" as a base material in road construction are summarized from the viewpoints of segregation from "Soil-like" and impurities, grading of "Concrete" and "Clay bricks", mechanical properties and environmental safety, and economic feasibility. The findings in this study raise challenges and perspectives to establish sound CDW management and to promote sustainable development of CDW recycling in Vietnam.

**Keywords:** construction and demolition waste (CDW); CDW landfills; waste composition; grain size distribution; Hanoi; Vietnam

## 1. Introduction

Unsustainable use of natural resources is a cause of the climate crisis and widespread environmental degradation (e.g., [1,2]). The construction industry is one of the major players that consumes natural resources as well as generates waste. Little attention, however,

has been paid to the saving of natural resources and material efficiency, and the waste generated by the construction industry (i.e., construction and demolition waste; CDW) is not fully recycled, and abundant natural resources have been used by the construction industry [3]. CDW accounts for a large amount of total waste generated all over the world, and the quantity and the composition of CDW vary between regions and depend on region-specific factors such as population, construction activities and materials, and traditions [3]. For example, it was reported that the total amount of generated waste was estimated to reach 30–40% in China [4], >30% in Australia [5], and 34% in Europe [6].

Southeast Asia countries have been experiencing remarkable growth in the construction industry, and the numbers of construction and demolition projects have been increasing rapidly due to urbanization and population growth. It was reported that the construction industry contributed approximately 5% of the total value added in Southeast Asia countries; however, construction is one of the main sources of energy and natural resource consumption [7]. It was also reported that approximately 40% of input materials of the global economy are used for construction activities, and unrestrained disposal of CDW becomes a serious waste of finite natural resources [8,9]. Hoang et al. [10] reviewed CDW management in Southeast Asian countries and suggested a need for more holistic and aggressive methods to achieve sustainable CDW management, including the development of legalized systematic approaches to CDW data collection and databases, a public private partnership for establishing recycling facilities, and internalization of informal actors in a formal CDW management arrangement. From the viewpoints of material efficiency and conservation of finite natural resources, more attention has been paid to sound CDW management and recycling to achieve sustainable development in developing countries, e.g., [11,12].

With rapid urbanization and economic growth, the generation of CDW has increased rapidly in Vietnam. Especially in large cities in Vietnam such as Hanoi, Hai Phong, Da Nang, and Ho Chi Minh City, the daily waste generation in Vietnam has reached approximately 6000 tons, in which the CDW accounts for 10–12% of total solid waste according to the Ministry of Natural Resources and Environment (MONRE) [13]. In Hanoi, it was reported that the daily CDW generation was projected to increase from 1500 in 2011 to 4000 tons in 2020 [14], and the annual increment in the generation of building demolition waste was estimated to be 4–5% from 2016 to 2020 [15]. The construction industry, on the other hand, is one of major sectors of hazardous waste production in Vietnam. Nguyen [16] reported that the industrial hazardous waste from construction industry accounted for 23.5% of total hazardous waste, including corrosive, toxic, and combustible materials.

Currently, the generated CDW is not fully recycled in Vietnam, and a major disposal method of generated CDW is dumping at designated sites. The collected CDW is landfilled without any treatment or recycling (hereafter, the CDW dumping sites are called 'CDW landfills'). For example, approximately 40–50% of CDW generated daily is estimated to be brought to CDW landfills in Hanoi [17]. In addition, illegal dumping of CDW is found frequently at roadsides and drainage canals in both urban and rural areas of the city [18], creating risks to human health and the environment, including soil and groundwater contamination, air pollution, transportation obstacles (traffic accidents), degradation of the urban landscape, and economic loss [19,20].

To implement better and sound CDW management, various policies and regulations that promote reuse and recycling of CDW were implemented in Vietnam [21,22]. The amended 'National Strategy for Integrated Solid Waste Management to 2025' with a vision towards 2050 [23,24] set a target of 90% for CDW collection and 60% for reuse or recycling of CDW. Circular No. 8/2017/TT-BX-D [25] on Construction Solid Waste Management formulated legal procedures such as separation, storage, transportation, reuse, recycling, and disposal of CDW, responsibilities of stakeholders, reporting forms, and a database of CDW management of responsible agencies. To develop proper CDW management and promote reuse and recycling of CDW in Vietnam, further challenges and opportunities are highly expected [21].

Currently, concrete and clay brick waste generated from construction and demolition sites is commonly recycled and used for various civil engineering purposes in developed countries (e.g., [26,27]). In developing countries including Vietnam, on the other hand, unrecyclable and/or hardly recyclable materials as well as hazardous and toxic materials are mixed with concrete and clay brick waste and dumped together at CDW landfills due to the lack of waste segregation. As a result, the insufficient waste segregation generates mixed construction/demolition waste and becomes one of the obstacles to promoting recycling and reuse (e.g., [28,29]). Until now, many investigations were carried out to characterize CDW generated from construction and building demolishing sites (e.g., [30,31]), but very limited studies and no reliable data and information on CDW landfills such as present management practices and conditions or waste composition of CDW dumped at landfills are available in Vietnam. To discuss suitable and applicable treatment technologies for recycling dumped materials at CDW landfills and to examine potential uses of recyclable and reusable materials and their feasibility for civil engineering purposes, it is necessary to identify the actual condition of CDW landfills and to characterize dumped CDW at landfills.

This paper, therefore, aims (i) to identify the actual conditions of CDW landfills under operation and closed/abandoned in Hanoi through a data collection survey, and (ii) to characterize the waste composition and the grain size distribution of CDW dumped at landfills. Finally, this paper summarizes challenges and recommendations obtained through the survey results and discusses the potential availability of dumped CDW as a base material in road construction.

## 2. Methodologies

### 2.1. Basic Information Survey of CDW Landfills

The survey for collecting basic information of CDW landfills in Hanoi, Vietnam was carried out in 2018–2019. A total of seven CDW landfills were investigated: Van Noi, Dan Phuong, Nguyen Khe, Vinh Quynh, Duong Lieu, Thanh Tri, and Nhat Tan. The locations of investigated CDW landfills are shown in Figure 1. To identify the actual condition of CDW landfills, information and data were collected by interviews of landowners and operating companies, searches were made on the internet, and site visits were carried out. A survey sheet is shown in Table 1. In the table, the survey result from the Thanh Tri CDW landfill is exemplified. The survey sheet consists of five items: (1) site information, (2) dumped waste and site condition, (3) surrounding environment, (4) photos and sketches, and (5) others.

**Table 1.** Survey form for collecting basic information on CDW landfill.

Date:          Surveyor:          Informant:

| 1. Site information | | |
|---|---|---|
| 1 | Name of landfill site | Thanh Tri |
| 2 | GPS location | 20° 59′ 21.6″ N, 105° 53′ 58.1″ E |
| 3 | Estimated area (m²) | 25,000 |
| 4 | Landowner | Hanoi People's Committee (public land) |
| 5 | Operation company/institute | Waste Treatment & Investment for Development of Hanoi Environment Joint Stock Company |
| 6 | Number of workers | 4 guards |
| 7 | Intake per day/week/month | Unknown (fluctuates) |
| 8 | Operation years | From 2017 to present |
| 9 | Estimated life in years | 5 years (up to 2022) |
| 10 | Groundwater level | 5-10 m from the ground surface |
| 11 | Acceptable waste | CDW (brick, concrete, tile, stone, wood, glass, plastic, steel, soil) |
| 12 | Previous land use | Vacant land (Free area formed by sedimentation of Red River) |
| 13 | Height of dumped waste | ~10 m in height |
| 14 | Surrounding environment | Red River, agricultural field, river-sand stock company |
| 15 | Accessibility | Good (20–30 min drive from center of Hanoi) |
| 16 | Workability | Enough space to work |
| 17 | Climate | Wet season: May–October, heavy rain Dry season: November–April, little rain |

| 2. Dumped waste and site condition | | |
|---|---|---|
| 1 | Dumped waste | ⊠Concrete   ⊠Brick   ⊠Tile   ⊠Rock   ⊠Wood ⊠Plastic   ⊠Steel   ⊠Soil □Others (                    ) |
| 2 | Burnable waste | ⊠No □ Yes (                              ) |
| 3 | Hazardous waste | □Oil □PCB □Asbestos □Incinerated ash □Others (                              ) |
| 4 | Depth of dumped waste | ⊠<4 m □4–10 m □>10 m |
| 5 | Covering soil | □Whole □Partial   ⊠None |
| 6 | Facility type | □Intermediate treatment   ⊠Final disposal □Transfer station □Illegal dumping |
| 7 | Incinerator | ⊠No □ Yes (                              ) |
| 8 | Fire damage in the past | ⊠No □ Yes (                              ) |
| 9 | Odor | □Whole   □Partial   ⊠None If odor: □Hydrogen sulfide □Enteruria □Disinfectant   □Oil   □Smoke □Others (                              ) |
| 10 | Collapses and caves | ⊠No □ Yes (                              ) |
| 11 | Vegetation | □Normal □Abnormal □Discolored □Unnaturally dry   ⊠No vegetation |
| 12 | Insects | □None □Centipedes   ⊠Flies □Mosquitoes □Other (                              ) |
| 13 | Discolored soil | ⊠No □ Yes (                              ) |
| 14 | Discolored water | ⊠No □ Yes (                              ) |

Table 1. *Cont.*

| 3. Surrounding environment | | |
|---|---|---|
| 1 | Water environment | ☐None ☒River ☐Pond ☐Channel |
| 2 | Well | ☐No ☒Yes (5–6) m depth from ground surface |
| 3 | Collapse/Caves | ☒ No ☐ Yes ( ) |
| 4 | Odor | ☐No ☒ Yes (Rotten smell ) |
| 5 | Insects | ☐None ☐Centipedes ☒Flies ☐Mosquitoes ☐Other ( ) |
| 6 | Intruding water | ☒No ☐ Yes ( ) |
| 7 | Discolored soil | ☒No ☐ Yes ( ) |
| 8 | Discolored water | ☒No ☐ Yes ( ) |
| 9 | Scattered | ☒No ☐ Yes ( ) |

**4. Photos and sketch**

Point ① 

Point ② 

Point ③ 

Point ④

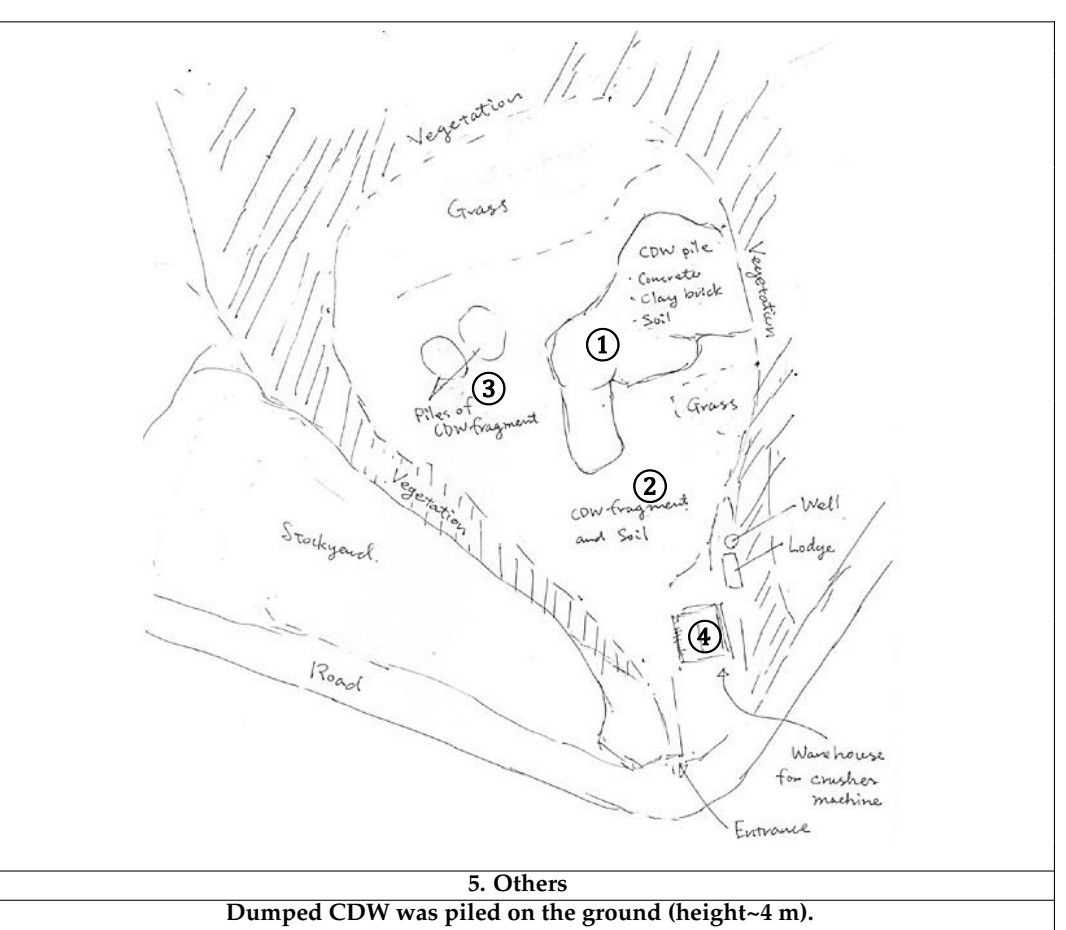

**5. Others**

Dumped CDW was piled on the ground (height~4 m).
Stock yards of construction materials such as crushed stone and sand were located next to the disposal site.
A mobile type crushing machine is stored in the site but currently the machine is used infrequently.
Daily intake fluctuates greatly with a maximum of 10 trucks.

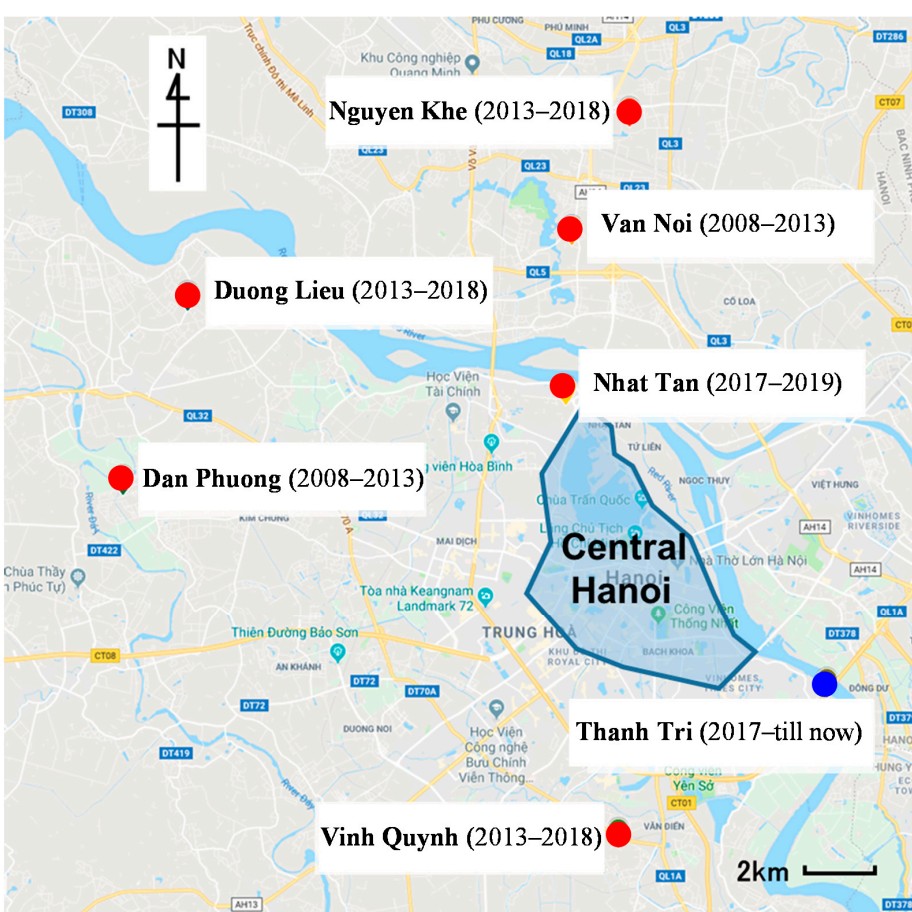

**Figure 1.** Locations of CDW landfills in Hanoi, Vietnam (Map data: © 2021 Google).

### 2.2. Waste Composition Survey at CDW Landfill

Among the seven CDW landfills at which basic information was collected, the waste composition survey was carried out at two CDW landfills, Thanh Tri and Vinh Quynh. A field survey manual for illegal waste dumping sites [32] was fully referenced to develop the methodology. In addition, a pit sampling [33–35] based on the judgement sampling design [30] was applied to decide the location and the size of the sampling pit at each CDW landfill, fully considering the results of a pre-survey, experience of experts and professionals, safety of workers, and limitations of time and money. For the pit sampling, a volume of 7.6 m$^3$ with a depth of 0.85 m (corresponding to a total mass on a wet basis of ~1.10 tons) was excavated at the Thanh Tri CDW landfill, and a volume of 6.6 m$^3$ with a depth of 1.27 m (corresponding total mass on wet basis is ~1.24 tons) was excavated at the Vinh Quynh CDW landfill.

The overall procedure of the waste composition survey is shown in Figure 2. The survey consisted of a total of six steps. Step 1 was the "Preliminary survey" including a site visit to check the site condition and to collect site information through an interview of the site manager. Step 2 was "Planning" to decide the location of the excavation pit and the working space, excavation and sampling methods, sorting materials, grading of material sizes, working schedule, and safety measures at the site. Step 3 was the "Preparation" that is necessary for the survey, such as obtaining equipment and materials for excavation, sorting, weighing, sampling, recording, and safety measures. The preparation of a working station and the recruitment of human resources were also organized. Step 4 was "Implementation" of the survey on the site, including excavation, sorting, and grading of materials, weighing of sorted materials, and bag sampling. Step 5 was "Laboratory testing". The water contents were measured to determine the dry weight of sorted materials. Step 6 was "Data analysis and reporting". By analyzing the collected data, the composition of sorted materials and

the grain size distributions of dumped waste materials can be summarized in a report. Details of the survey methodology are available from the JST-JICA SATREPS report [36].

**Step 1: Preliminary survey**
- Site visit (site condition)
- Site information (interview questionnaire)
- Trial excavation (if necessary)

**Step 2: Planning**
- Location of excavation pit and working space
- Excavation and sampling method
- Determination of sorting of materials and grading of material size
- Working schedule
- Safety measures

**Step 3: Preparation**
- Equipment and items (excavation, sorting, weighting, bag sampling, recording, safety measures)
- Preparation of working station
- Recruitment of human resources

**Step 4: Implementation**
- Excavation
- Sorting of materials and grading of material size
- Weighting sorted materials (wet mass)
- Bag sampling (for laboratory tests)
- Backfilling of excavated pit and cleaning

**Step 5: Laboratory testing**
- Water content measurement (convert to dry mass)
- Other measurements (physical and chemical properties, water and acid extractable ions, and so on)

**Step 6: Data analysis and reporting**
- Water composition of sorted materials (wet or dry mass basis, total and each size range)
- Particle size distribution of sorted materials (total and each material)
- Analysis of other measurements

**Figure 2.** Flow chart of overall procedure of waste composition survey.

### 2.3. Waste Classification and Material Sorting of Dumped Materials

CDW consists of materials such as concrete, clay bricks, asphalt pavement, wood, plastic, ceramics, and other materials generated during construction, renovation, and demolition of buildings and structures. In general, not only bulky materials but also stones and soil-like materials (debris) generated during construction and demolition works are commonly dumped together at CDW landfill sites. In addition, the waste generated at construction and demolition sites contains hazardous wastes such as asbestos, solvents, coal tar, and so on. Due to the lack of segregation, hazardous waste is often dumped at CDW landfill sites together with non-hazardous CDW in developing countries [37]. The definition of CDW also varies depending on country, region, and local conditions such as construction materials and structural styles [38–40]. The classification and the category of hazardous waste also vary depending on country, region, and its hazardous properties. For reference, the definition, the classification, and the remarks on CDW management including separation, reuse, and recycling shown in legal documents and technical standards of Vietnam are summarized in the Appendix A (Table A1).

On the other hand, a number of classification systems for municipal solid waste were proposed based on waste type, material groups, degradability of organic compounds, and so on [41,42]. Based on previous classification systems, typical dumped materials at CDW landfills in Vietnam are classified as shown in Table 2. The dumped CDW consists of non-hazardous and hazardous waste, and the non-hazardous waste is classified as inorganic and organic. The inorganic materials consist of non-degradable ones such as concrete, clay bricks, ceramics, glass, stones, soil-like, and degradable (corrosive) materials such as metals. The organic materials are further classified into putrescible and non-putrescible, but most non-hazardous CDW is non-putrescible (e.g., wood, plastic, paper). Except for

the soil-like material, the size of non-hazardous CDW at CDW landfill sites is normally >10 mm based on the results from the basic information survey. The waste composition survey in this study, therefore, classified the excavated materials (Step 4 in Figure 2) into ten categories: (1) concrete, 2) clay bricks, (3) ceramics, (4) glass, (5) plastics, (6) metals, (7) wood, (8) stones, (9) miscellaneous (including paper, rubber, leather, and textiles), and (10) soil-like (typically < 10 mm). Sorted materials in the survey are shown in Figure 3.

**Table 2.** Classification of materials dumped at CDW landfill.

| [Non-hazardous waste] | | | | |
|---|---|---|---|---|
| | **Inorganic** | | **Organic** | |
| Size | (Degradable) | (Non-degradable) | (Putrescible) Readily biodegradable | (Non-putrescible) Slowly biodegradable |
| ≥10 mm | ● Metals (corrosive) | ● Concrete ● Clay bricks ● Ceramics ● Glass Asphalt ● Stones | | ● Wood ● Plastic Paper Rubber Leather Textiles  } ● Miscellaneous |
| <10 mm | | ● Sol-like | | |
| [Hazardous waste] | | | | |
| | Gypsum (hazardous under reducing conditions) | Asbestos (fibers) Glass fibers Mercury (volatiles) Waste acid, Waste alkali | Paint (volatile) Detergents Solvents (volatile) | Oil Lubricants Coal tar (hazardous) |

●: Materials sorted in the waste composition survey. Notes: "Concrete" includes concrete and cement blocks. "Clay bricks" includes mortar binders. "Ceramics" includes tiles made of ceramics. "Plastics" includes soft plastics (e.g., PET bottles) and hard plastics (e.g., PVC pipes). "Metals" includes steel and copper cables. "Soil-like" includes fine fractions such as sludge, ash, and so on.

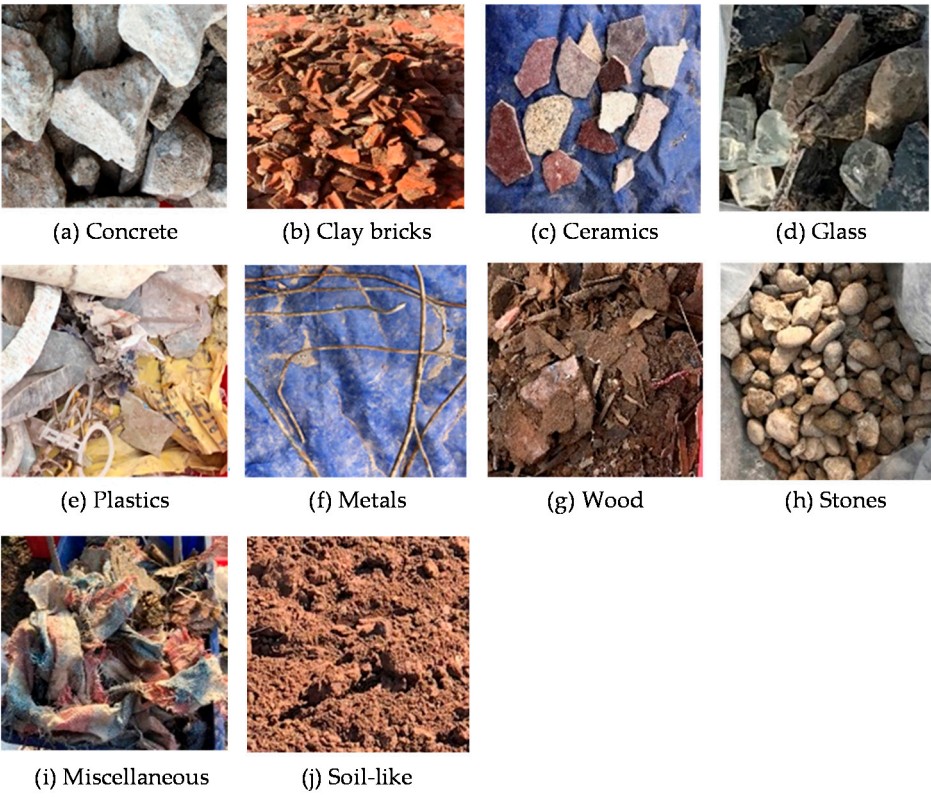

(a) Concrete     (b) Clay bricks     (c) Ceramics     (d) Glass

(e) Plastics     (f) Metals     (g) Wood     (h) Stones

(i) Miscellaneous     (j) Soil-like

**Figure 3.** Photos of materials sorted in the waste composition survey.

As well as sorting by material, the excavated materials were graded by size for investigating the grain size distribution of dumped CDW. A total of eight sieve sizes, >300, 106–300, 53–106, 26.5–53, 9.5–26.5, 4.75–9.5, 2–4.75, and <2 mm, were applied. The fractions <106 mm were measured by mechanical sieving with a set of sieves (fractions of 53–106, 26.5–53, 9.5–26.5 mm are labeled "Over 10", and fractions of 4.75–9.5, 2–4.75, <2 mm are "Under 10"), and the fractions >106 mm (labeled "Over 100") were measured by using mechanical sieving and an image analysis technique with photos [43–45]. For image analyses, the samples were placed on a tarpaulin as a flat plane and photographed horizontally on-site, and photos of samples were taken from the top of the placed samples (around 1.5 to 2 m height) as vertically as possible to avoid image distortion. The photo image was analyzed using image processing software (Adobe Illustrator, Adobe Inc., San Jose, CA, USA) to correct the distortion of images, and ImageJ (an open-source image processing program) was used to calculate the grain size. After correcting the distortion of the image, it was first binarized with a threshold value that shows a clear edge of each sample. Then, best-fit ellipse major axis, minor axis, and Feret's diameter (maximum caliper) were calculated automatically, and the volume and the equivalent diameter of each sample were estimated from the measured ellipsoid. Next, the estimated volume was converted to weight by multiplying it by the density of the material. Finally, the dataset consisting of the estimated weight and the equivalent diameter (grain size) was used to plot the grain size distribution. An illustration of overall scheme for material sorting and grading (Step 4 in Figure 2) is shown in Figure 4.

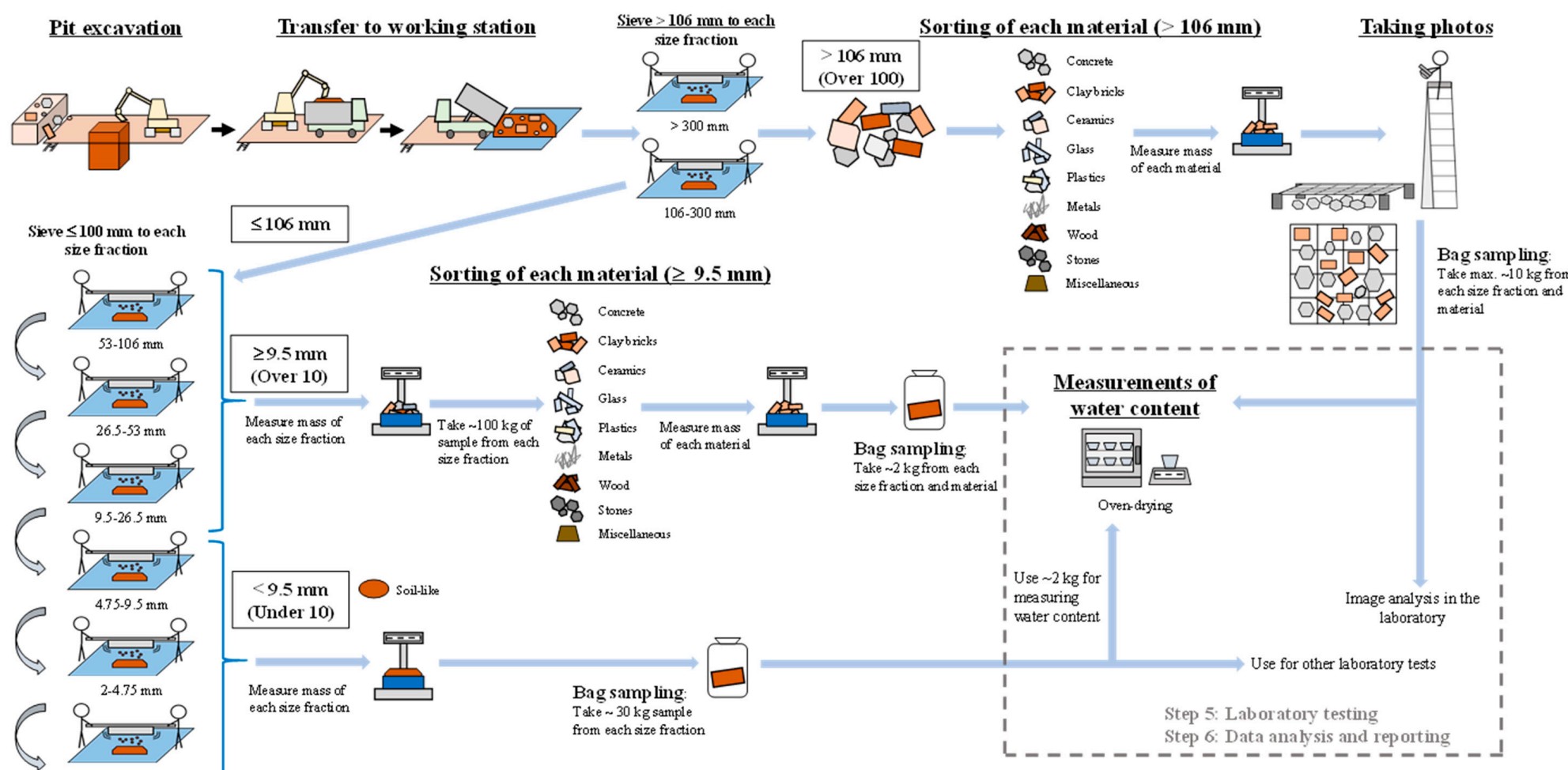

**Figure 4.** Overall scheme for material sorting and grading (Step 4 in Figure 2).

## 3. Results and Discussion

### 3.1. Characteristics of CDW Landfills

Results of the basic information survey are summarized in Table 3. Together with the comments of interviewees, characteristics of CDW landfills in Hanoi are observed as follows:

1.  Landowner and operating company: All investigated sites are public land belonging to the Hanoi People's Committee and operated by appointed companies under management of the Department of Construction (DOC) except for the Nhat Tan site (which is an illegal dumping site).
2.  Land area: The land areas of the six closed/abandoned CDW landfills and one operating landfill (Thanh Tri) varied widely between 2000 to 94,000 m$^2$.
3.  Operation status: Five official CDW landfills (Van Noi, Dan Phuong, Nguyen Khe, Vinh Quynh, and Duong Lieu) and one illegal landfill (Nhat Tan) are either closed or abandoned. Of the five official landfills, the Hanoi People's Committee was scheduled to change the land usage purpose of four landfills. Currently, the Thanh Tri landfill (an active landfill) accepts a limited amount of CDW because the site is planned to host a pilot CDW recycling line (although the implementation is not clearly decided; this was an interview answer from an operation company).
4.  Dumped waste: All CDW landfills investigated accepted/accept mainly CDW such as concrete and clay brick waste, soil, and sludge generated from construction and demolition works. The dumped CDW was mostly piled on the ground (4–10 m in height) with a shallow excavation of ~1 m. Small or moderate amounts of domestic waste were seen in all landfills. The Duong Lieu site, however, is the worst, with a large amount of domestic waste dumped on the whole ground surface, releasing a rotten smell.
5.  Previous land use: Ponds, abandoned agricultural land, and vacant lots were used to develop CDW landfills in Hanoi. An officer of an operation company stated that "Official CDW landfills originated from existing illegal ones in the past. The authority simply announced the land to be an official CDW landfill site and appointed (our) company to operate the site. Besides, it is true that proper site management is difficult".

In addition, some observations were found through the basic information survey and interviews to operation companies:

6.  Boundary of site: Due to the absence of fences and guards, the boundary between the site (dumping area) and the surrounding land was not clear under operation except for Thanh Tri CDW landfill, meaning that it was easy to enter the site (abandoned/closed landfills) without any hindrance.
7.  CDW treatment facility: One mobile type of crushing machine is available only at Than Tri CDW landfill, but the machine is not frequently used at present due to a lack of demand for crushing the CDW brought to the site.
8.  Manuals for site management and recording documents: No manuals for site management and documents that record the amount of accepted CDW and site inspection were found. In addition, the site workers did not receive sufficient training for managing the site properly.

### 3.2. Water Contents of Materials Sorted by Grain Size

A record sheet of the waste composition survey at Thanh Tri CDW landfill is shown in Table 4. The mass (in kg) recorded on the sheet is given on a wet basis (actual mass in kg at the site). After the measurement of water content in the laboratory (Step 5 in Figure 2), the measured mass of each grain size and sorted material was converted to the mass on dry basis. The measured water contents of sorted materials by each grain size as well as percentage of dry mass at Thanh Tri and Vinh Quynh CDW landfills are summarized in Table 5.

**Table 3.** Summary of basic information survey of CDW landfills.

| | CDW Landfill | Van Noi | Dan Phuong | Nguyen Khe | Vinh Quynh | Duong Lieu | Thanh Tri | Nhat Tan |
|---|---|---|---|---|---|---|---|---|
| 1 | Operation years | 6 years (2008–2013) | 6 years (2008–2013) | 6 years (2013–2018) | 6 years (2013–2018) | 6 years (2013–2018) | 2017 to present (up to 2022) | 3 years (2017–2019) |
| 2 | Landowner | Hanoi People's Committee | Hanoi People's Committee | Hanoi People's Committee | Hanoi People's Committee | Hanoi People's Committee | Hanoi People's Committee | Public land |
| 3 | Estimated land area | 94,000 m$^2$ | 37,000 m$^2$ | 7000 m$^2$ | 44,000 m$^2$ | 2000 m$^2$ | 25,000 m$^2$ | 25,000 m$^2$ |
| 4 | Previous land use | Unknown | Abandoned paddy field | Pond | Pond | Abandoned agricultural land | Vacant land | Abandoned agricultural land |
| 5 | Dumped waste | CDW, domestic waste (partially) | CDW | CDW, soil and sludge | CDW, soil and sludge, domestic waste (partially) | CDW, soil, domestic waste (partially) | CDW, soil | CDW, soil and sludge |
| 6 | Daily intake | 40–50 trucks | 30–40 trucks | 10–40 trucks | 20–30 trucks | 10–20 trucks | <10 trucks (fluctuates) | 40–50 trucks |
| 7 | Height and/or depth of dumped waste | Unknown * | Unknown * | ~5 m in height | ~4 m in height ~1 m in depth | <1 m in height | ~10 m in height ~1 m in depth | ~10 m in height |
| 8 | Surrounding environment | Van Tri Lake, agricultural land, golf course | Residential areas (under development) | Clay brick plant, pond, driving school | Concrete plant, irrigation channels, agricultural land | Agricultural land, residential houses | Red River, agricultural land, river sand stockyard | Concrete plant, high-rise buildings |
| 9 | Groundwater level (depth from ground surface) | Unknown ** | Unknown ** | ~10 m | ~3 m | Unknown ** | ~5 m | ~4 m |
| 10 | Other remarks | The land was developed after closure by Hanoi People's committee and is currently used for a concrete plant. | The land is scheduled to be developed as a public park from 2018 by Hanoi People's committee. | Most piled CDW was removed, and the land is used as a stockyard of clay brick plant. An excavated ground hole at the central zone. | The land is scheduled to be developed from 2019 by Hanoi People's committee. | Scattering of dumped domestic waste and rotten smell. The land is scheduled to develop from 2019 by Hanoi People's committee. | Under operation with 2 guards. Slight rotten smell from surrounding area. | Illegal dumping at night. |

* The land was altered since closing. ** The groundwater level could not be estimated due to the absence of a well inside/surrounding the site.

**Table 4.** Record sheet for sorted and graded materials in the waste composition survey at Thanh Tri CDW landfill.

| Mass of each size fraction after sieving | | |
|---|---|---|
| No. | Sieve size (mm) | Mass (kg) |
| 1 | >300 | 55 + 62 + 31 = 148 |
| 2 | 106–300 | 254 + 273 + 233 + 289 + 308 + 466 = 1823 |
| 3 | 53–106 | 271 + 295 + 291 + 321 + 279 + 303 + 303 = 2061 |
| 4 | 26.5–53 | 359 + 383 + 385 + 371 + 245 = 1743 |
| 5 | 9.5–26.5 | 517 + 485 + 489 + 493 + 679 = 2663 |
| 6 | 4.75–9.5 | 177 + 296 + 358 = 831 |
| 7 | 2–4.75 | 122 + 221 + 289 = 632 |
| 8 | < 2 | 347 + 258 + 223 + 230 = 1058 |

| Mass of each material for each size fraction | | | | | |
|---|---|---|---|---|---|
| Grain size: >300 mm | | Mass (kg) | Grain size: 26.5–53 mm | | Mass (kg) |
| 1 | Concrete | 10.2 + 6.4 = 16.6 | 1 | Concrete | 179 + 109 + 125.2 = 413.2 |
| 2 | Clay bricks | 57 + 30 + 44.4 = 131.4 | 2 | Clay bricks | 303 + 288 + 211.8 + 357 = 1159.8 |
| 3 | Ceramics | – | 3 | Ceramics | 99 + 60.9 = 159.9 |
| 4 | Glass | – | 4 | Glass | – |
| 5 | Plastics | – | 5 | Plastics | – |
| 6 | Metals | – | 6 | Metals | – |
| 7 | Wood | – | 7 | Wood | – |
| 8 | Stones | – | 8 | Stones | – |
| 9 | Miscellaneous | – | 9 | Miscellaneous | 10 |
| Grain size: 106–300 mm | | Mass (kg) | Grain size: 9.5–26.5 mm | | Mass (kg) |
| 1 | Concrete | 233 + 189.2 + 327 + 166.9 = 915.9 | 1 | Concrete | 279 + 301 + 288 + 206 + 259.7 = 1333.7 |
| 2 | Clay bricks | 177 + 247 + 180.5 + 226.8 = 831.3 | 2 | Clay bricks | 59 + 104 + 388.1 = 461.1 |
| 3 | Ceramics | 4.3 | 3 | Ceramics | 40.7 + 49 = 89.7 |
| 4 | Glass | – | 4 | Glass | 20.8 |
| 5 | Plastics | – | 5 | Plastics | – |
| 6 | Metals | – | 6 | Metals | – |
| 7 | Wood | – | 7 | Wood | 9.9 |
| 8 | Stones | 6.8 | 8 | Stones | 53 + 29.3 = 82.3 |
| 9 | Miscellaneous | 32.5 + 32.3 = 64.8 | 9 | Miscellaneous | 177 + 208 + 280.4 = 665.4 |
| Grain size: 53–106 mm | | Mass (kg) | | | |
| 1 | Concrete | 350 + 377 + 289 + 435.5 = 1451.5 | | | |
| 2 | Clay bricks | 156 + 294.2 = 450.2 | | | |
| 3 | Ceramics | 50 + 78.3 = 128.3 | | | |
| 4 | Glass | – | | | |
| 5 | Plastics | – | | | |
| 6 | Metals | – | | | |
| 7 | Wood | – | | | |
| 8 | Stones | – | | | |
| 9 | Miscellaneous | 31.3 | | | |

**Table 5.** Water contents of sorted materials in each grain size at Thanh Tri and Vinh Quynh CDW landfills.

| Grain Size (mm) | Material | Thanh Tri CDW Landfill | | Vinh Quynh CDW Landfill | |
|---|---|---|---|---|---|
| | | Water Content (%) | % of Dry Mass | Water Content (%) | % of Dry Mass |
| >300 | Concrete | 2.3 | 11.3 | 6.0 | 97.9 |
| | Clay bricks | 4.9 | 88.7 | 10.5 | 2.1 |
| | Ceramics | – | – | – | – |
| | Glass | – | – | – | – |
| | Plastics | – | – | – | – |
| | Metals | – | – | – | – |
| | Wood | – | – | – | – |
| | Stones | – | – | – | – |
| | Miscellaneous | – | – | – | – |
| | Total | 100 | Total | 100 | |
| 106–300 | Concrete | 2.5 | 51.2 | 8.2 | 73.2 |
| | Clay bricks | 5.9 | 44.8 | 12.0 | 10.3 |
| | Ceramics | 6.6 | 0.2 | 0.5 | 2.3 |
| | Glass | – | – | – | – |
| | Plastics | – | – | – | – |
| | Metals | – | – | – | – |
| | Wood | – | – | – | – |
| | Stones | 2.7 | 0.3 | 1.9 | 0.7 |
| | Miscellaneous | 8.9 | 3.4 | 40.0 | 13.5 |
| | Total | 100 | Total | 100 | |
| 53–106 | Concrete | 4.6 | 71.0 | 9.4 | 64.3 |
| | Clay bricks | 6.9 | 21.5 | 13.3 | 19.6 |
| | Ceramics | 7.6 | 6.1 | 0.5 | 7.7 |
| | Glass | – | – | – | – |
| | Plastics | – | – | – | – |
| | Metals | – | – | – | – |
| | Wood | – | – | 84.9 | 0.2 |
| | Stones | – | – | 2.0 | 2.0 |
| | Miscellaneous | 9.2 | 1.4 | 46.2 | 6.2 |
| | Total | 100 | Total | 100 | |
| 26.5–53 | Concrete | 8.8 | 23.5 | 9.2 | 66.0 |
| | Clay bricks | 7.5 | 66.8 | 16.1 | 13.6 |
| | Ceramics | 7.6 | 9.2 | – | – |
| | Glass | – | – | 0.3 | 0.1 |
| | Plastics | – | – | – | – |
| | Metals | – | – | – | – |
| | Wood | – | – | 111 | 0.0 |
| | Stones | – | – | 2.4 | 2.9 |
| | Miscellaneous | 17.5 | 0.5 | 35.8 | 17.4 |
| | Total | 100 | Total | 100 | |
| 9.5–26.5 | Concrete | 11.4 | 49.5 | 12.9 | 41.7 |
| | Clay bricks | 9.4 | 17.5 | 15.6 | 5.6 |
| | Ceramics | 5.1 | 3.6 | 0.3 | 1.0 |
| | Glass | 0.4 | 0.9 | 0.4 | 0.7 |
| | Plastics | – | – | – | – |
| | Metals | – | – | – | – |
| | Wood | 19.0 | 0.3 | – | – |
| | Stones | 2.8 | 3.4 | 2.0 | 8.4 |
| | Miscellaneous | 10.7 | 24.9 | 26.6 | 42.6 |
| | Total | 100 | Total | 100 | |
| 4.75–9.5 | | 16.8 | 100 | 26.4 | 100 |
| 2–4.75 | Soil-like | 27.8 | 100 | 26.5 | 100 |
| <2 | | 27.8 | 100 | 29.3 | 100 |

It can be observed that the water contents of "Concrete" and "Clay bricks" depended on the grain size. For "Concrete", the water content increased with decreasing grain size and ranged from 2.3% (>300 mm) to 11.4% (9.5–26.5 mm) at the Thanh Tri CDW landfill and ranged from 6.0% (>300 mm) to 12.9% (9.5–26.5mm) at the Vinh Quynh CDW landfill. Likewise, the water contents of "Clay bricks" increased with decreasing grain size and ranged from 4.9% (>300 mm) to 9.4% (9.5–26.5 mm) at the Thanh Tri CDW landfill and ranged from 10.5% (>300 mm) to 16.1% (26.5–53 mm) and 15.6% (9.5–26.5 mm) at the Vinh Quynh CDW landfill.

On the other hand, the water contents of "Ceramic", "Glass", and "Stones" did not vary depending on the grain size. The water contents of "Ceramic" ranged from 5.1–7.6% at Thanh Tri CDW landfill and 0.3–0.5% at Vinh Quynh CDW landfill. "Glass" materials were found in two fractions of 26.5–53 and 9.5–26.5 mm, and their water contents ranged 0.3–0.5%. The water contents of "Stones" ranged 1.9–2.8%. The water contents of "Wood" and "Miscellaneous" categories were higher than those of other sorted materials and were 84.9–111% for "Wood" and 6.2–46.2% for "Miscellaneous". "Soil-like" materials had a relatively small variation in the water and ranged from 16.8% to 29.3% irrespective of grain size.

### 3.3. Waste Composition of Dumped CDW

Waste compositions of the sorted material on a dry mass basis (percentage of dry mass) at Thanh Tri and Vinh CDW landfills given in Table 5 are shown in graphs of Figure 5. For both landfills, major components of dumped materials were "Concrete", "Clay bricks", and "Soil-like". The sum of these materials became 88.1% of the total at Thanh Tri CDW landfill and 82.9% of the total at Vinh Quynh CDW landfill. The secondary component was "Miscellaneous" at 7.1% at Thanh Tri and 13.1% at Vinh Quynh. Other materials were very small and varied from 0.02 (given as 0.0% in Figure 5) to 3.7% of the total. It is worth noting that hazardous and toxic materials were not found clearly through the survey at both CDW landfills.

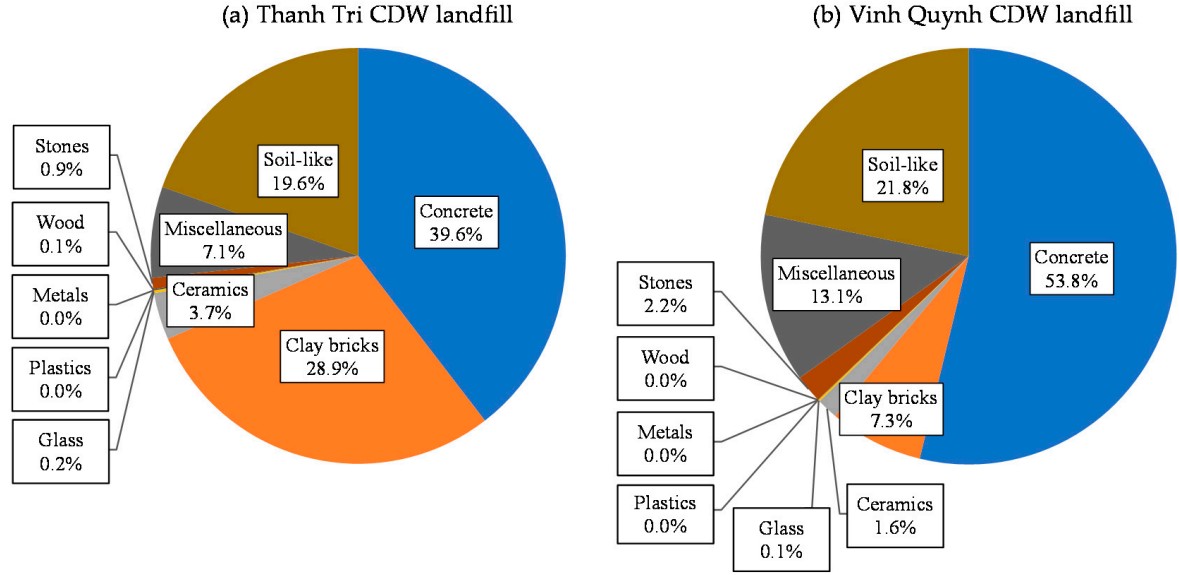

**Figure 5.** Waste composition of dumped CDW on dry mass basis: (**a**) Thanh Tri CDW landfill, (**b**) Vinh Quynh CDW landfill.

The measured waste compositions of dumped CDW are consistent with reported waste composition data that were measured at CDW generation sources in Vietnam. For example, Hanoi Urban Planning Institute (HUPI) [46] reported the composition of construction solid waste in Hanoi was 23% "Concrete", 31% "Brick", and 36% "Soil, sand, and gravel", and the sum of these materials was 90%. Hoang et al. [31] surveyed CDW compositions from five construction and 10 demolition sites and reported that the waste composition was 32%

"Concrete", 14% "Brick with mortar", and 42% "Soil", and the sum of these materials was 93%. Compared to those reported values, on the other hand, an important difference can be found in the percentages of "Metal". Both HUPI [46] and Hoang et al. [31] reported that "Metal" was 5% of the total generation, while "Metal" of dumped CDW in this survey was 0.0% for both Thanh Tri and Vinh Quynh CDW landfills. This suggests that such a valuable material was fully segregated and recovered on-site before being brought to CDW landfill sites.

### 3.4. Grain Size Distribution of Dumped CDW

Measured grain size distributions of all sorted materials on both wet and dry mass bases are shown in Figure 6a (Thanh Tri CDW landfill) and Figure 6b (Vinh Quynh CDW landfill). For both landfills, the grain size was distributed broadly up to the maximum diameter of 600 mm. There was no significant difference in the grain size distributions between wet and dry masses.

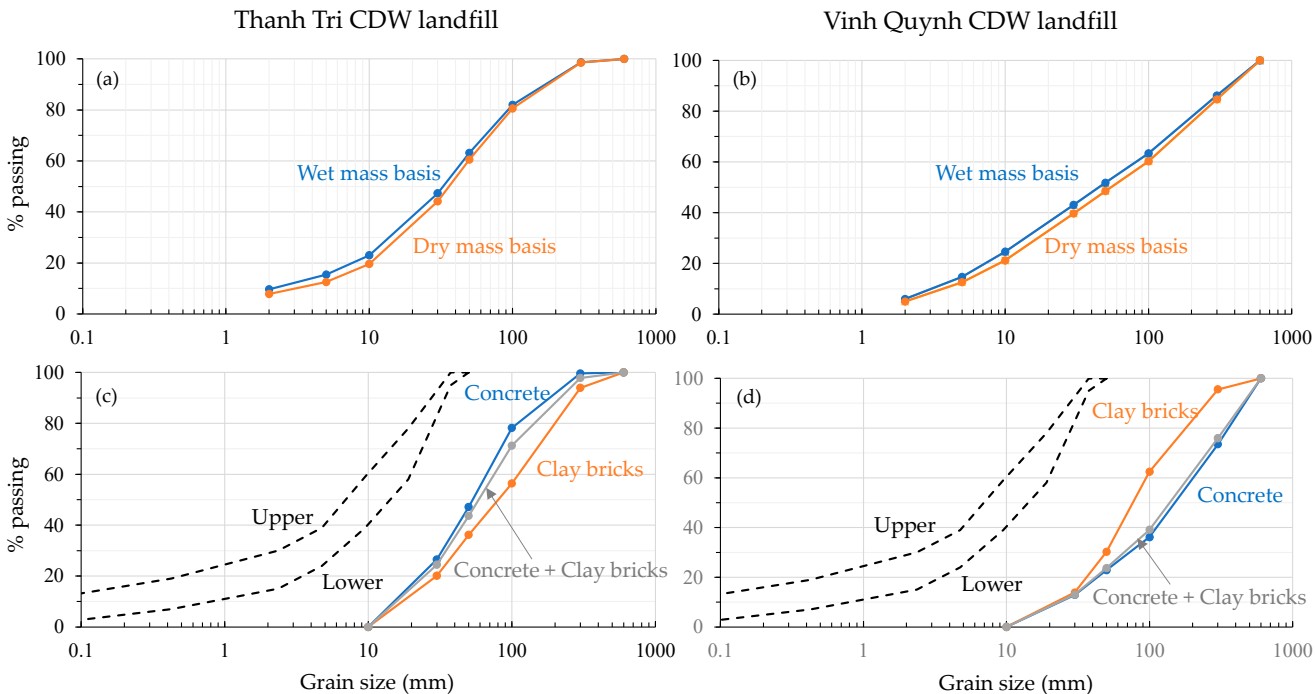

**Figure 6.** Grain size distributions of dumped CDW: all materials sorted on wet and dry mass bases (**a,b**), concrete, clay bricks, and concrete + clay bricks (≥10 mm) on dry mass basis (**c,d**). Upper and lower limits of grain sizes (Dmax = 37.5 mm) in TCVN 8859:2011 are shown by broken lines.

The values of dry mass weight percentage of "Concrete" and "Clay bricks" (≥10 mm) were extracted from all sorted materials, and their grain size distributions are shown in Figure 6c,d. The grain size distributions of "Concrete + Clay bricks" are also shown in the figures. The grain size distribution of "Clay bricks" was similar between two CDW landfills, and 50% of diameter ($D_{50}$) was 84 mm for Thanh Tri and 81 mm for Vinh Quynh. On the other hand, there was a big difference in $D_{50}$ values of "Concrete" and "Concrete + Clay bricks" in two CDW landfills, giving $D_{50}$ = 55–61 mm for Thanh Tri and $D_{50}$ = 159–174 mm for Vinh Quynh. This indicates that more crushed and/or finer fractions of concrete waste were brought to the Thanh Tri CDW landfill from construction and demolition sites because the crushing of concrete waste was not done on-site at either landfill site.

### 3.5. Relationship between Sorted Materials and Grain Size of Dumped CDW

The relationship between dry mass distribution and grain size of sorted materials is shown in Figure 7. At the Thanh Tri CDW landfill, the dry mass of "Concrete" and

"Clay bricks" was distributed well, ranging between 9.5–300 mm. At the Vinh Quynh CDW landfill, on the other hand, two clear peaks of "Concrete" sizes can be seen within the ranges of 106–300 and >300 mm, indicating that coarser fractions of concrete were dumped there compared to the Thanh Tri CDW landfill. The dry mass of "Clay bricks" was much smaller than that of "Concrete" at the Vinh Quynh CDW landfill. For both landfills, "Miscellaneous" was mostly in the range of 9.5–26.5 mm. In addition, a relatively high dry mass of ceramics within the range of 9.5–106 mm was found at the Thanh Tri CDW landfill, but very little ceramic material was found at the Vinh Quynh CDW landfill.

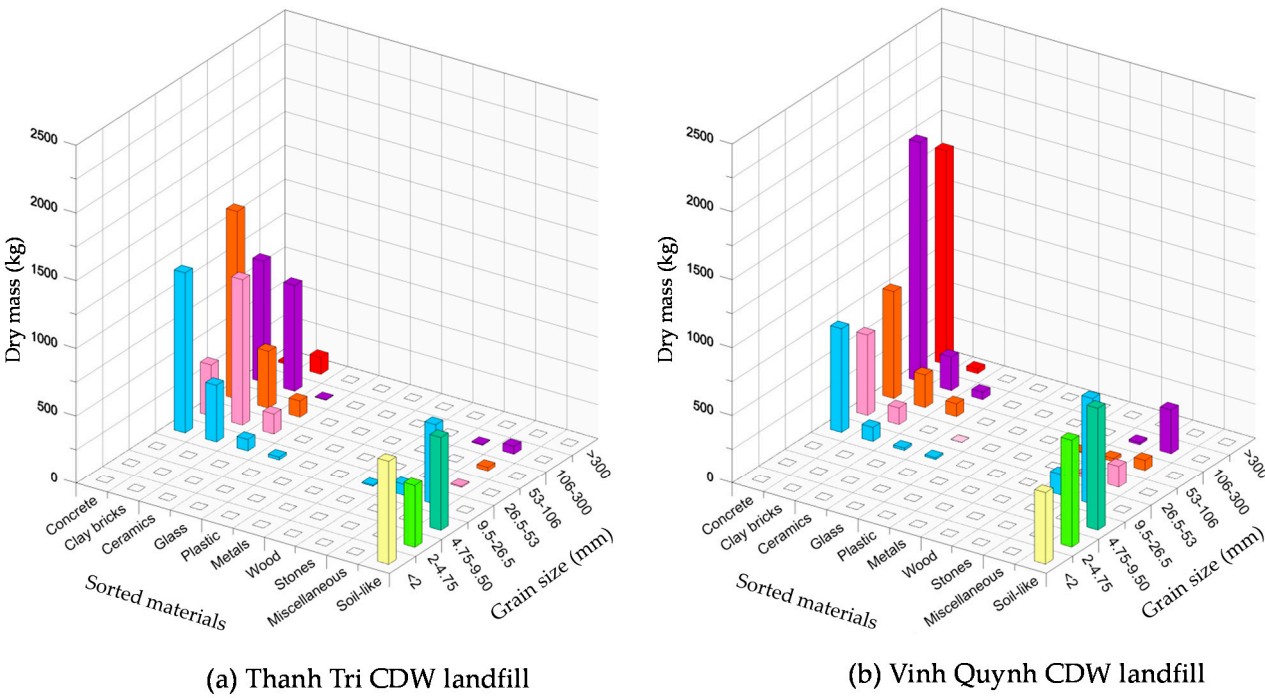

(a) Thanh Tri CDW landfill  (b) Vinh Quynh CDW landfill

**Figure 7.** Relationship between dry mass distribution of sorted materials and grain size: (**a**) Thanh Tri CDW landfill, (**b**) Vinh Quynh CDW landfill.

## 4. Challenges and Perspectives

### 4.1. Management and Development of CDW Landfill

Based on the results of the basic information survey and interviews of operating companies of CDW landfills, it was found that the land area of CDW landfills varied widely from 2000 to 94,000 m$^2$ while the operation period was mostly 6 years. As of January 2021, only one landfill site (Thanh Tri) was under operation in Hanoi. The site is still able to accept CDW, but the remaining capacity cannot be expected to last a long time without promoting appropriate treatments of accepted/dumped CDW, e.g., waste separation, volume reduction by crushing, and recycling of concrete and clay brick waste. Except for the Nhat Tan landfill, the sanitary condition of landfills was relatively good with normal vegetation, no insects, no discolored soil/water, and no hazardous waste. For other sites, however, more or less illegal dumping of domestic waste by surrounding residents could be observed. The main reason for illegal dumping of domestic waste is supposed to be the lack of a clear boundary between dumping and non-dumping areas, especially at abandoned/closed landfill sites. That allows the residents to enter easily. The other problem is that there is no recorded document issued by a responsible authority even though the site was declared as an official CDW landfill. Rather, there are solely verbal proof and/or reports in the newspaper. Moreover, CDW landfill sites are not fully equipped for treating accepted/dumped CDW.

For improving the current management of CDW landfills and for promoting recycling of CDW, challenges and recommendations are summarized as follows:

1. Record of CDW landfill sites:

A suitable recording system that describes the types and the amounts of accepted CDW, the generating/discharge places of construction and demolition works, and a designated CDW transportation company is necessary to manage CDW landfill sites in an environmentally-sound manner and estimate the duration of operation (i.e., lifetime).

2. Technical support for management of CDW landfill sites:

Training of staff members and development of technical guidelines are necessary for proper management and operation of CDW landfill sites and to minimize environmental impacts on surrounding areas (including the restraint of illegal dumping of domestic waste inside sites).

3. Plan and strategy for developing and renovating CDW landfill sites:

A proper plan and strategy for developing CDW landfill sites must consider fully the source generation (e.g., scheduled construction and development plans, planned demolition works), transportation of generated CDW, and surrounding environment. In addition, not only new development but also renovation of abandoned/closed CDW landfills should be considered to extend the lifetime of CDW landfills to increase the confidence in the CDW landfilling process*.

* Currently, the Hanoi master plan on solid waste management [41] depends highly on the establishment of new CDW landfill sites, zoning urban and rural areas to achieve the CDW collection and recycling percentage shown in National Strategy for Integrated Solid Waste Management to 2025 with vision towards 2050 [18,19]. Hanoi City, however, is facing serious difficulty obtaining suitable land for developing new waste landfill sites due to technical and financial restrictions and complaints from residents surrounding the planned land.

4. Promotion of CDW recycling:

Along with dumping the CDW in a safe manner, alternative methods to treat CDW (e.g., reuse and recycling CDW) must be considered, studied, and implemented. The recycling and reuse of dumped CDW also contribute to extending the lifetime of existing CDW landfill sites. Besides, the quality of recycled materials from CDW must be improved to gain the trust of users (e.g., construction industry).

5. Lessons learned from developed countries with high CDW recycling:

Many developed countries such as Japan, the Netherlands, Germany, and Denmark have made much effort to establish sound CDW management and to promote the CDW recycling and reduction of CDW emission in recent years, and those countries achieved recycling of >80% of CDW [11,12]. For example, in Japan, the clients and the contractors must have an agreement that obligates material sorting and recycling activities, and the contractors must submit detailed CDW recycling procedures before starting demolition and construction activities according to Construction Material Recycling Law [39]. Those lessons and approaches should be incorporated to upgrade the CDW management and the management of CDW landfill sites in Vietnam step by step by adjusting the social structure and the system, including the relationships among authorities, private sectors, and public in Vietnam.

Moreover, the CDW management, including sound CDW landfill sites as well as the CDW recycling, can be assessed by SWOT (strengths, weaknesses, opportunities, and threats) analysis, e.g., [47–49], and reverse logistics, e.g., [50,51], and a proper plan-do-check-act (PDCA) cycle that is commonly used for waste reduction, e.g., [52–54], can contribute to promote the CDW management plan and strategy. When we adopt these analyses and assess the effectiveness of plan and strategy, first of all, it is important to clarify both drivers/advantages and barriers/disadvantages of the current CDW situation,

fully considering the actual material flow and the monetary flow of CDW in target area. Especially, the role and the contribution of informal sectors/actors involved in CDW collection and disposal shall be investigated as well as the recycling trade hierarchy among the sectors/actors [10,55].

*4.2. Potential Availability of Dumped CDW as a Base Material in Road Construction*

The waste composition survey in this study showed that the major components of dumped CDW were "Concrete", "Clay bricks", and "Soil-like", and the sum of these materials reached >80% of the total (Figure 5). The measured grain size distributions (Figures 6 and 7) showed that "Concrete" ranged between 10–600 mm with $D_{50}$ = 55–61 mm at Thanh Tri and D50 = 159–174 mm for Vinh Quynh. "Clay bricks" ranged mainly between 10–300 mm with D50 = ~80 mm. Based on the observations, technical recommendations to examine potential availability of dumped CDW for civil engineering purposes, especially as a base material in road construction, are summarized as follows:

1.   Segregation of "Soil-like" and impurities:

As a first step, the separation of "Soil-like" from "Concrete" and "Clay bricks" is necessary. By separating with a 9.5 mm sieve, ~20% of "Soil-like" (Under 10) can be separated (Figure 5). After the separation, the fractions of "Over 10" and "Over 100" still contain "Ceramic", "Glass", "Stones", "Miscellaneous", and small amounts of "Plastics", "Wood", and "Metals" as impurities. Because the amounts of impurities reach ~12% of the total at Thanh Tri and ~17% of the total at Vinh Quynh, the impurities should be carefully excluded from "Concrete" and "Clay bricks". Additionally, the separation of "Concrete" and "Clay bricks" is recommended especially for the "Over 100" fraction for securing the subsequent quality control of recycled aggregates.

2.   Grading of "Concrete" and "Clay bricks":

Next, "Concrete" and "Clay bricks" of "Over 10" and "Over 100" categories are crushed and graded using a crusher with a suitable sieve machine (e.g., apertures of sieve: 5, 10, and 40 mm). In Vietnam, the grading of aggregates for roadbed materials is described in TCVN 8859 [36]. The upper and the lower limits of grain sizes with the maximum diameter (Dmax) of 37.5 mm are given in Figure 6c,d. As shown in the figures, the measured grain sizes of "Concrete" and "Clay bricks" are larger than those in TCVN 8859 [56]. For manufacturing the aggregates, not only $D_{max}$ but also grain size indices such as 50%, 25%, and 75% of diameters in grain size distribution must be controlled by adjusting the specifications of the crusher (e.g., feed amount, crusher speed, operation torque).

3.   Mechanical properties and environmental safety of graded aggregates:

After crushing and grading, the mechanical properties and the environmental safety of recycled aggregates should be examined. The technical standards and requirements for recycled aggregates for roadbed materials have not yet been developed (as of January 2021), but the requirements in TCVN 8859 [56] such as the bearing ratio (CBR $\geq$ 100% at K $\geq$ 98% where CBR is the California bearing ratio and K is the degree of compaction), plasticity (wL $\leq$ 25% and IP $\leq$ 6 where wL is the liquid limit and IP is the plastic index), and durability on impact (Los Angeles abrasion $\leq$35%) can be possible indicators to evaluate physical and mechanical properties of recycled aggregates. For the environmental safety of recycled aggregates, Vietnam National Technical Regulations such as QCVN 07:2009/BTNMT [57] and QCVN 03-MT:2015/BTNMT [58] should be considered. Moreover, the mixing of clay brick aggregates with concrete aggregates as well as chemical and mineralogical compositions should be examined. Due to their vulnerability and fragility, the mixing of clay brick aggregates affects the compaction and the bearing capacities of recycled aggregates as roadbed materials. For example, compaction densities decreased mostly with increased proportions of clay brick aggregates [59,60], decreasing the bearing capacity [61]. The allowable limit of clay brick aggregates (and other types of aggregates such as reclaimed asphalt pavement and recycled glass) mixed with concrete aggregates

should be examined fully when we apply the recycled aggregates made from dumped CDW to road construction as roadbed materials [62].

4. Economic feasibility:

Economic profitability is an important trigger to promote CDW recycling. Many studies evaluated the economic feasibility and the applicable treatment technologies based on the cost–benefit analysis and characterized a monetary flow of CDW recycling [63–65]. To cut costs and determine the investment value of recycling dumped CDW as a roadbed material, good quality data on initial and operating costs are needed as well as accurate projections of CDW generation and construction demand and the market/procurement price of recycled materials [66].

## 5. Conclusions

This paper described the present management conditions of CDW landfills in Hanoi, Vietnam and characterized the waste composition of CDW dumped at landfills through basic information and waste composition surveys. Currently, only one landfill site is under operation in Hanoi. The site is still able to accept CDW; however, the remaining capacity cannot be expected to last long. Thus, plans and strategies for developing and renovating CDW landfill sites must be implemented in the near future. The sanitary condition of investigated landfills was relatively good and lacked dumping of hazardous waste. On the other hand, illegal dumping of domestic waste by residents was observed more or less at all sites due to an unclear boundary between dumping and surrounding areas. To improve current management of CDW landfills, a suitable recording system of accepted/dumped CDW and technical support for site managers are required. Along with proper management of CDW landfills, an alternative solution is to treat CDW, e.g., encourage CDW recycling. Based on the results of waste composition surveys at two CDW landfills, the major components of dumped CDW were "Concrete", "Clay bricks", and "Soil-like", and the sum of these materials was >80% of the total. Grain size distributions showed that "Concrete" ranged between 10–600 mm, and "Clay bricks" ranged between 10–300 mm in size. Technical recommendations to examine the potential availability of dumped "Concrete" and "Clay bricks" as a base material in the road construction are summarized from the viewpoints of segregation of "Soil-like" and impurities, grading of "Concrete" and "Clay bricks", mechanical properties and environmental safety, and economic feasibility. Further research and laboratory tests are needed to support the proposed technical recommendations; however, the challenges and the perspectives suggested in this paper, including the increase of the confidence in the CDW landfilling process and the trust of recycled materials from CDW landfill sites, would contribute to the improvement of the management of CDW landfill sites in Hanoi. Moreover, the present management system of CDW should be assessed by SWOT analysis and reverse logistics, and it is essential to introduce a proper PDCA cycle for assessing sound CDW management and promoting sustainable development of CDW recycling in Vietnam.

**Author Contributions:** Conceptualization, H.G.N., Y.I. and K.K.; methodology, D.T.N. and M.K.; software, A.K. and A.M.; validation, H.G.N., Y.I. and K.K.; formal analysis, A.K., A.M. and K.K.; investigation, resources, and data collection, D.T.N., H.T.N., V.C.T., A.K. and A.M.; writing—original draft preparation, H.G.N.; writing—review and editing, K.K.; visualization, H.G.N. and K.K.; supervision, Y.I., M.K. and K.K.; project administration, H.G.N. and K.K.; funding acquisition, H.G.N. and K.K. All authors have read and agreed to the published version of the manuscript.

**Funding:** This research was supported by JST–JICA Science and Technology Research Partnership for Sustainable Development Program (SATREPS) project (No. JPMJSA1701).

**Institutional Review Board Statement:** Not applicable.

**Informed Consent Statement:** Not applicable.

**Data Availability Statement:** The data presented in this study are available on request from the corresponding author. The data are not publicly available due to the information security conditions of the project.

**Acknowledgments:** We thank the Ministry of Construction (MOC) in Vietnam and Hanoi Department of Construction (DOC), and Hanoi Urban Environment Company (URENCO) for their support to the field survey. We acknowledge Tselmeg Aldarjav, Ngo Thanh Ha, Hao Ningning, former master's degree students of Saitama University, for their dedicated effort on field work and laboratory tests.

**Conflicts of Interest:** The authors declare no conflict of interest. The funders had no role in the design of the study; in the collection, analyses, or interpretation of data; in the writing of the manuscript, or in the decision to publish the results.

## Appendix A

**Table A1.** Definition, classification, and remarks on CDW management in legal documents and technical standards of Vietnam.

| TCVN/Decree/Circular | Definition | Classification | Remarks on CDW Management |
|---|---|---|---|
| Law on Urban Planning [67] | None | None | The Law prescribes urban planning activities including elaborating, evaluating, approving, and adjusting urban planning; organizing the implementation of urban planning and managing urban development according to approved urban planning.<br>The Law provides that the requirements of urban planning shall meet the needs for technical infrastructures, including waste treatment, and that the solid waste treatment planning must indicate the total volume of solid waste, locations, and sizes of transfer depots, solid waste treatment facilities, auxiliary works and sanitation distance from solid waste treatment facilities. |
| Law on Construction [68] | None | None | The Law prescribes the rights, obligations, and responsibilities of agencies, organizations, individuals, and state management in construction investment activities.<br>The Law provides that construction contractors shall take measures to ensure construction wastes are collected and treated in a proper manner and to protect the environment in the course of construction including solid waste prescribed by the Law on environmental protection. |
| Law on Environmental Protection [69] | None | None | The Law prescribes statutory provisions on environmental protection activities; measures and resources used for the purpose of environmental protection; rights, powers, duties and obligations of regulatory bodies, agencies, organizations, households and individuals who are tasked with the environmental protection task.<br>The Law provides that waste treatment works must be included in the construction design and budget of manufacturing and business establishments that produce wastes that negatively impact the environment and that solid wastes and other wastes are collected and treated in accordance with environmental standards. |
| Law on Public Investment [70] | None | None | The Law prescribes the management and use of the capital budget for public investment; the state management of public investment; the right, obligation and responsibility of agencies, organizations and individuals involved in public investment activities.<br>The Law provides that the community shall supervise compliance with the regulations of the law on investment, construction land, waste treatment and environment protection. |

**Table A1.** *Cont.*

| TCVN/Decree/Circular | Definition | Classification | Remarks on CDW Management |
|---|---|---|---|
| Decree No.59/2007/ND-CP on solid waste management [71] | Not specified but general definition of solid waste is given: Solid waste means waste in a solid form, discarded from production, business, service, daily life, or other activities. Solid waste includes ordinary solid waste and hazardous waste. | (a) Soil and sludge from excavation and dredging of surface soil layer (b) Soil, stone, solid waste from construction materials (bricks, roofing titles, mortar, concrete and discarded adhesive materials) (c) Solid waste such as broken glass, discarded iron and steel, dead wood, paper and plastic bags | (a) Construction solid waste shall be separated (b) Soil and sludge can be used to fertilize vegetation soil. (c) Soil, stone, solid waste from construction materials (bricks, roofing titles, mortar, concrete, and discarded adhesive materials) are recyclable or reusable as filling materials for construction works. (d) Solid waste such as broken glass, discarded iron and steel, dead wood, paper, and plastic bags are recyclable and reusable. (e) Hazardous solid waste shall be separated at the source and stored separately according to regulations and not mixed with ordinary solid waste. Mixed hazardous and ordinary solid waste shall be disposed of like hazardous solid waste. |
| Decree No.38/2015/ND-CP on management of wastes and scraps [72] | Solid waste from construction activities (including renovation and demolition of works) | (a) Soil, sludge (b) Gravelly soil, solid waste from construction materials (brick, tile, grout, concrete, adhesive materials) (c) Recyclable solid waste such as glass, steel, wood, paper, plastics | (a) Soil, sludge from excavation, dredging topsoil, and digging foundation piles shall be used to cultivate crop land or suitable land areas. (b) Gravelly soil, solid waste from construction materials (brick, tile, grout, concrete, adhesive materials) shall be recycled as construction materials or reused as backfill materials for buildings or buried in construction solid waste landfill. (c) Recyclable solid waste such as glass, steel, wood, paper, and plastics shall be recycled and reused. |
| Decree No.59/2015/ND-CP on construction project management [73] | None | None | The Decree prescribes the responsibility of building contractor on the environment protection measures within and out of the construction site including anti-dust, noise control, waste treatment and construction site cleanup, and measures of safety and environmental hygiene on the delivery/transport of waste to regulated places. |

**Table A1.** *Cont.*

| TCVN/Decree/Circular | Definition | Classification | Remarks on CDW Management |
|---|---|---|---|
| Joint Circular No. 01/2001/TTLT-BKHCNMT-BXD on guiding the regulations on environmental protection for selection of the location for the construction and operation of solid waste burial sites [74] | Not specified but a general definition of solid waste is given: Solid wastes mean the solid wastes arising from daily-life activities in urban areas and industrial zones, which include wastes from population quarters, trade activities, urban services, hospitals, industrial wastes, and construction activities | None | The joint Circular guides the regulations on environmental protection for the selection of location for the construction and operation of solid waste burial sites. |
| Circular No. 01/2011/TT-BXD on guiding the strategic environmental assessment in construction and urban plans [75] | None | None | The Circular guides the strategic environmental assessment in construction and urban plans including identification of main environmental issues and solid waste treatment planning. |
| Circular No. 36/2015/TT-BTNMT on management of hazardous wastes: Appendix 1 [76] | None | Construction and demolition wastes (including excavated soil from contaminated sites) according to European list of wastes (Commission Decision 2000/532/EC 2000 [77]) | Technical requirements and management procedures for reuse, recycling, treatment are given in Appendix 2 of the circular. |

<div align="center">**Table A1.** *Cont.*</div>

| TCVN/Decree/Circular | Definition | Classification | Remarks on CDW Management |
|---|---|---|---|
| Circular No. 08/2017/TT-BXD on construction solid waste management [25] | Construction solid waste (CSW) is solid waste generated during surveying and construction works (including new construction, renovation, improvement, rehabilitation, recovery, and demolit ion) | (a) Soil<br>(b) Concrete<br>(c) Asphalt and concrete<br>(d) Sludge<br>(e) Wood, paper<br>(f) Iron, steel<br>(g) Others | (a) Concrete and brick debris can be recycled to produce coarse aggregates, to manufacture brick, wall panel, floor brick, other building materials, and leveling the ground surface.<br>(b) Wood and paper can be recycled mainly to produce paper, wood, and burnable materials.<br>(c) Mixtures of bitumen can be recycled to produce bituminous concrete (aggregate form).<br>(d) Steel and other metal can be reused or used for metal production.<br>(e) Wastes that cannot be recycled or reused are required to be landfilled.<br>(f) Hazardous waste is separated and managed following Decree No. 38/2015/ND-CP and other guiding legal documents on hazardous waste management |
| Circular No. 02/2018/TT-BXD on providing for environmental protection in construction and reporting thereof [78] | None | None | The Circular prescribes the responsibilities of project owners and building contractors for environmental protection during the implementation of construction works and reporting thereof. Reporting on environmental protection includes the preparation and provision of information, figures and data about sources of waste, impact of waste on the environment, waste management, and environmental protection in construction field. |
| TCVN 6705: 2009 on normal solid wastes—Classification [79] | Waste from construction activities: Waste discarded during dismantling or renovating old construction works, or new works in process of construction (house, bridge, road, etc.). | (a) Mortar<br>(b) Broken brick<br>(c) Concrete<br>(d) Ceramic<br>(e) Water pipeline<br>(f) Roof<br>(g) Gypsum<br>(h) Other materials | None |

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
