# Peer review of "Current Management Condition and Waste Composition Characteristics of Construction and Demolition Waste Landfills in Hanoi of Vietnam"

_sustainability, doi:10.3390/su131810148_

Round 1
Reviewer 1 Report
The authors present a manuscript for demolition wastes in Hanoi Vietnam.
The manuscript even though is quite informative does not provide any novelty in the field. It is more like a case study that is presented and not a research work.
The authors have limited related literature and required to deliver a strong solid action plan for the area.
They must connect their approach with the sustainability with the SDG Goals of UN and present synergies with other countries like Vietnam.
These are the basics. After the authors will revised their work I will return with some additional ones but minor considering these ones already mentioned.
Reviewer 2 Report
The paper does not provide any new scientific results.
The authors are asked to perform experiments on the complete characterization of construction and demolition waste landfills and to find ways to process these wastes.

Author Response
Thanks for reviewing our paper manuscript.
1. No new scientific results
The comment is unclear to us. Please let us see which points are necessary to improve and launch the scientific viewpoints. Besides, if the work, methodology, and results shown in the manuscript have been published already, please let us specify more details.
2. Experiments on the complete chracterization of CDW landfills
The given datasheet to characterize the CDW landfills has been carefully prepared by considering the long experience of illegal waste dumping survey in Japan and other countries, fully considering the site-specific (Vietnam) condition. If the survey sheet is insufficient, please point out how to improve it and let us clarify the what kinds of experiments are necessary to characterize the CDW landfills.
Reviewer 3 Report
The paper is interesting. Some comments can be found below:
-We live now in a climate emergency so its most strange that the authors have not start the paper by mentioning exactly that. It seems that they are not aware about the words of a Professor of Physics at the University of Oxford authored a paper where one can read the following:
“Let’s get this on the table right away, without mincing words. With regard to the climate crisis, yes, it’s time to panic”
Pierrehumbert, R., 2019. There is no Plan B for dealing with the climate crisis. Bulletin of the Atomic Scientists, pp.1-7.
So please start the introduction by draw a connection between environmental degradation, resource efficiency, and CDW.
- Focusing only in Vietnam right after line 3 is not a good option. Check the publication Pacheco-Torgal, F., Ding, Y.; Colangelo, F.; Tuladhar, R. Koutamanis, A. (2020) Advances in Construction and Demolition Waste Recycling. Elsevier
https://www.elsevier.com/books/advances-in-construction-and-demolition-waste-recycling/pacheco-torgal/978-0-12-819055-5
Especially Part 1 on Management Construction and Demolition Wastes
- The text at lines 58-66 is a duplicate of the text at lines 67-75
“Currently, the generated CDW is not fully recycled in Vietnam, and a major disposal method of generated CDW is dumping at designated sites. The collected CDW is land- filled without any treatment or recycling (hereafter, the CDW dumping sites are called ‘CDW landfills’). For example, approximately 40–50% of CDW generated daily is esti-61 mated to be brought to CDW landfills in Hanoi [6]. In addition, illegal dumping of CDW is found frequently at roadsides and drainage canals in both urban and rural areas of the city [7], creating risks to the human health and the environment, including soil and groundwater contamination, and air pollution, transportation obstacles (traffic accidents), degradation of the urban landscape, and economic loss [8,9]”
Author Response
Thanks for your valuable suggestions and comments.
- As suggested, we added the recommended paper and cited to UNEP website to emphasize the environmental degradation, material efficiency linking to CDW recycling in the first part of paper.
- Yes, we cited the recommended book chapter and added more info on the CDW generations in not only South Asian countries including Vietnam but also other countries.
- Thanks for pointing out the mistake. We have corrected it.
Round 2
Reviewer 1 Report
The following issues must be addressed by the authors:
- A SWOT analysis is required for the methods presented.
- A critical presentation of the sustainability related issues of the on-going methods in the demolition wastes, considering in addition the actual materials and resources lost.
- Comparison with the practices used in other countries/areas with the same charactiristics
- Comparison with the best applied methods in handling these types of wastes and a proposed path to successfully implement these techniques and methods in this area.
- Lessons learned and legislative/organizational changes required to adopt the most sustainable methods used globally in demolition wastes handling.
- Language improvement by a native English speaking person.
Of course the case studies are accepted as paper types, but their submission requires significant effort and analysis of the case study by the authors.
Author Response
We appreciate your valuable comments to improve the paper. We can fully realize the importance of the points that you have suggested. As given in the answers to specific comments, we still face limitations to identify the present situation of CDW system in Hanoi including the management of CDW landfill sites and the role of informal actors in the system. To solve the limitations, we are now on-going survey and investigation to characterize the actual material and monetary flow of CDW in Hanoi. Then, we plan to do SWOT analysis as well as the analysis based on reverse logistics and plan to establish Hanoi CDW recycling promotion committee to make a proper and sustainable PDCA cycle of CDW management and recycling. The answers to specific comments are below:
1. A SWOT analysis is required for the methods presented.
We agree that SWOT analysis as well as reverse logistics for having a qualitative and quantitative assessment of CDW management of target site. In “Section 4.1 (Lines 548-560)”, we added the importance of those assessment. Till now, however, unknown factors (but with high influence) such as the role of informal actors on CDW management and awareness/attitude of stakeholders (local authorities, private sectors, public) in Hanoi City have not been fully analyzed yet (because this paper mainly targets the management of CDW landfill sites). In future, we plan to do those analyses fully collecting the actual and practical data/information (also added in “5. Conclusions” as a prospect).
2. A critical presentation of the sustainability related issues of the on-going methods in the demolition wastes, considering in addition the actual materials and resources lost.
In “1. Introduction (Lines 57-72)”, we added an information on the importance of material efficiency and save of finite natural resources.
3. Comparison with the practices used in other countries/areas with the same characteristics
A detail review on CDW management in other countries with the same characteristics (especially, in Southeast Asian countries) has been done in Hoang et al. (2021) [reference [10]; “1. Introduction (Lines 64-69)”]. As well, we also introduced an info on CDW generation in some typical countries citing published papers [references [4-6]; “1. Introduction (Lines 51-56)”].
4. Comparison with the best applied methods in handling these types of wastes and a proposed path to successfully implement these techniques and methods in this area.
As shown in the paper (references no. 14, 21), many donor-oriented projects and government-oriented projects have been done to improve the CDW management system and to promote the recycling. But, the implementation of the proposed techniques and methods have not been achieved successful outcomes and results in Vietnam. We added newly a challenge/perspective as “5. Lessons learned from developed countries with high CDW recycling” in Section 4.1 (Lines 549-559) and emphasized the importance of the upgrading of CDW management system and recycling.
5. Lessons learned and legislative/organizational changes required to adopt the most sustainable methods used globally in demolition wastes handling.
Linking to the answer to the above comment, we have many good lessons and practices in some developed countries with high CDW recycling (added in Lines 560-6570). As well as the introduction of those good lessons, we also added the importance of PDCA cycle to assess the effectiveness of plan and strategy in Conclusions (Lines 652-657).
6. Language improvement by a native English speaking person.
We had native English check two times for original manuscript and revised manuscript.
Reviewer 2 Report
Dear Authors,
Please improve the paper according to the observations from the previous review.
Author Response
Comment: Please improve the paper according to the observations from the previous review.
Thanks for the comment. Yes, we added more information on the importance of material efficiency and save of finite natural resources, and review of common problem on CDW management especially in Southeast Asian countries including Vietnam (1. Introduction). Besides, the importance of good lessons and practices in some developed countries with high CDW recycling was added in Section 4.1. In the same section, we also added the importance of adoption by adopting SWOT analysis and reverse logistics analysis for evaluating the of present CDW management system and recycling practices, and the introduction of PDCA cycle to assess the effectiveness of CDW management plan and strategy.
Round 3
Reviewer 1 Report
No more comments
Author Response
1. Comments and Suggestions for Authors: No more comments
Thanks a lot.
Reviewer 2 Report
The authors are asked to perform experiments on the complete characterization of construction and demolition waste landfills and to find ways to process these wastes.
The complete characterization (chemical, mineralogical ...) of construction and demolition waste landfills missing from the paper.
Author Response
Comments and Suggestions for Authors: The authors are asked to perform experiments on the complete characterization of construction and demolition waste landfills and to find ways to process these wastes. The complete characterization (chemical, mineralogical ...) of construction and demolition waste landfills missing from the paper.
Answer: Thanks again for your kind comments.
This paper mainly targets the current situation of CDW landfills and material composition of buried waste in Hanoi. We agree the importance of those chracteristics of dumped materials especially for material recovery and recycling of buried waste. We are now having intensive study to characterize those chemical and mineralogical properties in other task. We hope we can report those results soon.